# Beyond health: A machine learning analysis of structural barriers to school attainment in Somalia

Jibril Abdikadir Ali[1], Mukhtaar Axmed Cumar[1], Mustafe Khadar Abdi[2], Abdisalam Hassan Muse[3], Hodo Abdikarim[2,3]*

1 Faculty of Education, School of Postgraduate Studies and Research (SPGSR), Amoud University, Borama, Somaliland, 2 Faculty of Science and Humanities, School of Postgraduate Studies and Research (SPGSR), Amoud University, Borama, Somalia, 3 Research and Innovation Centre, Amoud University, Borama, Somaliland

* hodo.abdikarim@amoud.edu.so

## Abstract

In fragile states like Somalia, the link between poor health and educational exclusion is critical yet poorly understood. This study uses a novel machine learning approach to identify and rank the most significant barriers to school attendance. We analyzed nationally representative data from 10,511 children aged 6–18 in the 2022 Somalia Integrated Household Budget Survey (SIHBS). Ten supervised machine learning models were employed to predict school attendance, with the Random Forest model emerging as the top performer (AUC = 0.86). Contrary to conventional wisdom, our findings reveal a clear hierarchy of barriers where structural and demographic factors are the most powerful predictors. A child's age (non-attendance rises from 6% in 6–10 year-olds to 25% in 15–18 year-olds), geographic region (non-attendance reaches 30.5% in Middle Shabelle), and residence type (nomadic children face triple the non-attendance risk of their peers) were the dominant determinants. These factors significantly outweighed the direct predictive power of individual health status or household poverty. The results indicate that while health is important, educational exclusion in Somalia is more fundamentally driven by where a child lives, how old they are, and their family's mode of life. Policy interventions must therefore shift from broad, single-sector approaches to geographically-targeted, age-specific strategies. To effectively address the root causes of educational inequality, efforts should prioritize service delivery in marginalized regions and provide tailored support for nomadic and adolescent populations.

## Introduction

Health and education constitute the foundational pillars of human capital, engaging in a cycle of mutual reinforcement that shapes both individual prosperity and national

**Data availability statement:** The data underlying the findings of this study are third-party data from the Somalia National Bureau of Statistics (SNBS). The minimal dataset used to reach the conclusions in this manuscript is derived from the Somalia Integrated Household Budget Survey (SIHBS) 2022. The data are publicly available from the SNBS Microdata Catalog. Interested researchers can apply for access and download the data directly from SNBS (https://microdata.nbs.gov.so/index.php/catalog/59) and SIHBS (https://nbs.gov.so/somalia-integrated-household-budget-survey-sihbs-2022). The specific variables, including their measurement and categorization, used in this analysis are detailed in the paper. The methodological details provided in the Methods section of our manuscript, combined with these variables obtained from the source above, are sufficient to replicate the study findings in their entirety. The authors confirm they had no special access privileges to this data. Any researcher can obtain the data through the same application process on the official SNBS Microdata Catalog.

**Funding:** The author(s) received no specific funding for this work.

**Competing interests:** The authors have declared that no competing interests exist.

development. Globally, higher educational attainment is correlated with improved health outcomes, while poor health can significantly impede a child's ability to learn and achieve academic success [1,2]. This synergy is particularly critical, yet fragile, in low- and middle-income countries (LMICs), where systemic stressors frequently undermine progress in both sectors. [3] contend that understanding these systems is essential for effective intervention. In many developing regions, the connection between health systems and educational stability is severely tested by disease outbreaks, climate shocks, and conflict [4,5]. These crises expose and exacerbate existing inequalities. For instance, gender-related barriers, including early marriage and the burden of domestic labor, consistently hinder girls from completing their education across Sub-Saharan Africa [6,7]. Furthermore, the rise of the digital divide introduces an additional layer of exclusion, as a lack of access to technology further marginalizes the most vulnerable students [8]. These challenges are magnified in conflict-affected nations. In contexts such as Nigeria, socio-economic precarity and regional insecurity are primary drivers of educational exclusion [9].

Similarly, the weaponization of public services in Syria has devastated health professional education, disrupting human capital development for an entire generation [10,11]. Such systemic fragility creates a persistent gap between policy aspirations and the lived realities of households, where interconnected factors related to health, economics, and security combine to keep children out of school. This reality is starkly evident in Somalia, a nation grappling with the legacy of a three-decade-long conflict. The country's health system is one of the most fragmented globally, with less than 30% of the population having access to basic care [12]. This systemic collapse has precipitated a severe mental health crisis, widespread psychosocial trauma, and one of the highest rates of mental illness worldwide [13,14]. The humanitarian health and nutrition response is hampered by chronic shortages of resources, poor infrastructure, and pervasive insecurity, making it difficult to reach vulnerable populations [15,16]. For the millions of internally displaced persons (IDPs) in Somalia, these barriers are even more acute. Displaced women and children face formidable challenges in accessing maternal and child health services due to security risks, prohibitive costs, and traditional beliefs [17–19]. This environment of systemic collapse raises a critical question: what are the primary drivers of educational exclusion in Somalia? While the nexus between health and education is globally acknowledged, its mechanisms in a context of such profound and overlapping crises remain poorly quantified [7,20].

A notable research gap persists in identifying and ranking the most influential determinants of school non-attendance within this fragile context. Although previous studies have examined poverty determinants using traditional methodologies [21,22], there has been an absence of a comprehensive, national-level analysis that models the combined and relative impact of health, demographic, and structural factors on educational attainment. This study, "Beyond Health: A Machine Learning Analysis of Structural Barriers to School Attainment in Somalia," seeks to address this gap. Employing a novel machine learning approach on a nationally representative dataset, this research aims to identify and rank the key predictors of school attendance. The

study's objective is to transcend conventional assumptions and provide a data-driven understanding of the hierarchy of barriers, thereby offering actionable evidence for designing effective, targeted policies to enhance both health and education for Somali children.

## Literature review

Understanding the barriers to school attainment in a fragile context necessitates an integrated theoretical lens that can account for influences operating at multiple levels. Key concepts such as the "health-education nexus" and "structural barriers" are central to this inquiry. The health-education nexus describes the bidirectional relationship where health is a prerequisite for effective learning, and education is a key determinant of long-term health [2]. Structural barriers, in contrast, refer to systemic obstacles embedded in social, economic, and political systems that limit opportunities for certain groups [6,23]. Human Capital Theory provides a foundational paradigm, positing that health and education are interdependent investments essential for future productivity [24,25]. Within this model, poor health directly depreciates a child's educational capital. However, this theory often overlooks the complex pathways and contextual factors that mediate this relationship in settings of extreme fragility [26,27]. It fails to explain why, in some contexts, broader environmental forces may be more decisive than individual or household investments.

To address this limitation, Bronfenbrenner's Ecological Systems Theory offers a more nuanced, multi-level perspective. It conceptualizes child development as being shaped by a series of nested environmental systems, from the microsystem of the family to the macrosystem of cultural norms and public policy [28–30]. This framework is indispensable for analyzing contexts like Somalia, as it allows for the simultaneous examination of individual health status, household conditions, and the wider community and security environment [31,32]. Social equity theories provide a critical lens for understanding the disparities that emerge from these interacting systems. Intersectionality theory, in particular, explains how overlapping social identities related to gender, class, ethnicity, and displacement create unique and compounded forms of disadvantage [33,34]. This is crucial for analyzing Somalia, where the barriers faced by a girl from a poor, nomadic family are fundamentally different from those faced by an urban boy. A theoretical gap persists, however, in operationalizing these complex frameworks. While ecological and intersectional theories effectively map potential influences, they lack a method for quantifying the relative predictive power of these different systemic layers. The existing literature does not offer a clear, empirically-tested hierarchy of which barriers—individual, household, or structural—are most potent in driving educational exclusion in a fragile state.

Empirical research has consistently demonstrated the detrimental effects of poor health on educational outcomes. A systematic review has confirmed a strong association between chronic health conditions and lower academic attainment, establishing physical well-being as a direct determinant of a child's educational trajectory [35]. Other studies have illustrated how health data can inform targeted interventions to enhance educational performance [36–38]. Specific health-related barriers, such as inadequate Menstrual Health and Hygiene (MHH), have also been documented to cause school absenteeism among adolescent girls [39,40]. In addition to health, a substantial body of literature identifies socio-economic and structural factors as significant determinants of educational access. In Nigeria, regional insecurity and household poverty are key drivers of educational exclusion [9,41]. Children with disabilities encounter compounded barriers, including inaccessible infrastructure and social stigma, which are often intensified in low- and middle-income countries (LMICs) [28,42]. In Somalia, specifically, studies highlight the profound barriers faced by internally displaced persons (IDPs) in accessing essential services, including healthcare and, by extension, education [17,21]. Clan dynamics and gender inequality further shape the humanitarian response and access to resources [14,43,44]. Parental and household characteristics also serve as critical mediators. A mother's education level, for instance, significantly affects a child's development through enhanced nutrition and health-seeking behaviors [24,45–47].

Conversely, gender-based violence and adolescent pregnancy remain substantial obstacles to girls' education in many parts of the world [6,34,48]. While these factors are widely recognized, a significant empirical gap remains. Most studies

examine these barriers in isolation or within specific subpopulations. There is a lack of research that synthetically analyzes the combined and relative impact of health, household, geographic, and livelihood factors on school attainment at a national scale in a conflict-affected country. Consequently, it remains unclear which of these many documented barriers are the most powerful drivers of exclusion in a context of systemic fragility. The methodological landscape for studying the health-education nexus has primarily relied on evidence synthesis and traditional statistical techniques. Systematic reviews and meta-analyses are the gold standard for summarizing research on policies and interventions [49,50]. Many empirical studies use cross-sectional survey data analyzed with conventional methods like logistic regression to identify associated factors [22,51]. While these methods are valuable for identifying statistically significant relationships, they are often limited by pre-specified assumptions about linearity and can struggle to model complex, high-order interactions within large datasets.

The advent of machine learning (ML) signifies a methodological paradigm shift, offering robust tools for prediction and pattern recognition. An expanding corpus of research illustrates the application of ML in health promotion, behavioral change, and digital health [45,52,53]. Systematic reviews have documented the increasing utilization of ML to predict child and adolescent health outcomes, highlighting its potential to analyze complex datasets that surpass the capabilities of traditional statistical methods [54,55]. In the educational sciences, ML is regarded as a potent framework for prediction, particularly as large and complex datasets become more prevalent [22,41,56,57]. Despite its potential, the application of ML to multifaceted development challenges in fragile states remains a significant methodological gap. While some studies have initiated the use of ML to analyze social issues in Somalia, such as poverty determinants [22,58], its application to the nexus of health, education, and structural inequality at a national level is novel. Methodological reviews of ML in medicine have underscored challenges such as the necessity for rigorous evaluation and transparent reporting to ensure findings are clinically relevant and not merely artifacts of benchmark-chasing [59,60]. This study addresses the methodological gap by applying a suite of supervised ML algorithms to a nationally representative dataset from Somalia. This approach transcends the identification of simple correlations to construct a predictive model that can rank the determinants of school non-attendance, thereby providing a data-driven hierarchy of barriers to inform policy.

## Methods and material

### Data source and study setting

This study utilized data from the 2022 Somalia Integrated Household Budget Survey (SIHBS) is a landmark national survey conducted by the Somalia National Bureau of Statistics (SNBS) with World Bank support. It represents the first comprehensive household budget survey since 1985, covering 7,212 households across 17 regions (excluding Middle Juba due to insecurity). The survey employs a stratified sampling design, distinguishing urban, rural, and nomadic populations, with a 96% response rate. Data was collected digitally via tablets, capturing detailed information on demographics, household welfare, consumption patterns, and access to services. Key objectives include updating GDP and CPI metrics, measuring poverty, and establishing baselines for future socio-economic monitoring. The SIHBS is a critical tool for evidence-based policymaking, aligning with Somalia's National Development Plan (NDP9) and the Sustainable Development Goals (SDGs) [61,62].

### Study variables

The analysis leverages data from the 2022 Somalia Integrated Household Budget Survey (SIHBS) to examine key factors influencing education and health outcomes in Somalia. The selected variables were carefully chosen based on existing literature and relevance to welfare indicators, capturing both individual and household-level characteristics. The study focuses on children aged 6–18, given the critical role of education and health in human capital development. One of the primary variables of interest is school attendance, recoded as a binary indicator (0 for "No" and 1 for "Yes") to assess enrollment rates. Health-related variables include chronic illness, which identifies long-term health conditions, and illness

in the last month serving as a proxy for recent healthcare needs. To measure the impact of health on daily life, the study examines days unable to perform normal activities categorized into 0 days, 1–7 days, and 8–30 days. Additionally, access to medical treatment is analyzed, with missing values imputed to ensure data robustness.

The analysis incorporates several control variables to account for socio-economic, demographic, and household-level differences. Age of the child is grouped into three categories (6–10, 11–14, and 15–18 years) to reflect developmental stages, while sex, region and Residence assess gender, geographic and dwelling disparities. Household structure variables provide critical context children lived with their mothers and with fathers revealing caregiving disparities, and household size house hold size is categorized as Small (1–3 persons), Medium (4–6 persons), or Large (7+ persons) to capture resource allocation dynamics. Additional controls include IDP status for displacement vulnerabilities and poverty status (poor; 0/1) for economic influences. Together, these variables spanning individual characteristics, family environments, and systemic factors create a robust framework to identify barriers and opportunities in Somalia's education and health sectors, ensuring policy recommendations address both immediate needs and structural determinants.

## Statistical analysis

All analyses were conducted using Stata 17 and R Studio, employing a dual analytical framework that combined traditional statistical methods with a machine learning approach to provide both explanatory and predictive insights. The study began with conventional analyses in Stata 17 to establish baseline associations. The distribution of school attendance (the dependent variable) was evaluated through proportion tables with 95% confidence intervals. Cross-tabulations were used to examine associations with key health and demographic indicators. Bivariate relationships between school attainment and all independent variables were quantified using Pearson's $\chi^2$ tests. A Bonferroni-corrected alpha of $\alpha = 0.025$ was used to reduce the risk of Type I errors arising from multiple comparisons.

Complementing these associative methods, a predictive modeling approach was implemented in R Studio to identify the most powerful determinants of school attendance. This approach is increasingly recognized for its ability to handle complex, non-linear relationships often found in large-scale health and social science data [41,63]. We tested a diverse suite of ten supervised machine learning algorithms: Random Forest, XGBoost, Support Vector Machine (SVM), K-Nearest Neighbors (KNN), Neural Networks, Naïve Bayes, Decision Tree, Probit Regression, Logistic Regression, and Gradient Boosting Machine (GBM). These models were deliberately chosen to represent different algorithmic families (e.g., tree-based ensembles, linear models, instance-based learners) to ensure a robust and comprehensive assessment of predictive patterns [64]. The dataset was partitioned into an 80:20 training-test split to evaluate model performance on unseen data. This dual-methodology approach aligns with a growing precedent of using machine learning to analyze complex survey data in development contexts, including for Somalia [22].

## Data preprocessing and model training

Prior to model training, several preprocessing steps were taken to ensure data quality and suitability for machine learning algorithms. Categorical variables with a large number of levels, such as 'Region', were retained as they were hypothesized to be critical predictors. To handle missing data, which was minimal but present in the 'Received Medical Treatment' variable (<1% of cases), we employed k-Nearest Neighbors (k-NN) imputation with k=5. This method was chosen over simpler techniques like mean/mode imputation because it uses information from related variables to estimate missing values, preserving the underlying data structure more effectively. All categorical predictor variables (e.g., Region, Residence, Sex) were then one-hot encoded to convert them into a numerical format suitable for the algorithms. The models were trained on the 80% training set using default hyperparameters from the R caret package. This decision was made because the primary objective of this study was not to build a perfectly optimized model for deployment, but rather to compare the relative predictive power of different model families and identify the most important features [59]. This comparative approach allows for a robust identification of the key barriers to school attainment.

## Model evaluation and feature importance

Model performance was rigorously evaluated on the 20% holdout test set using a range of standard metrics to provide a holistic view of predictive capability. These metrics included the F1 score (the harmonic mean of precision and recall), balanced accuracy (to account for the imbalanced nature of the outcome variable), sensitivity (true positive rate), specificity (true negative rate), and the Area Under the Receiver Operating Characteristic Curve (AUC-ROC). The AUC is a particularly robust measure of a classifier's ability to distinguish between classes across all possible thresholds. The Random Forest model was ultimately selected for the final feature importance analysis. This choice was based on its superior overall performance, particularly its high AUC and specificity, and its well-established ability to generate interpretable feature importance scores without requiring linear assumptions [54]. Feature importance was calculated using the Mean Decrease in Gini metric, which measures how much each variable contributes to the homogeneity of the nodes and leaves in the resulting random forest. This method provides a clear and intuitive ranking of the variables based on their predictive power in the best-performing model.

## Results and discussion

### Descriptive statistics

**Prevalence of education attainment in Somalia.** As shown in Table 1, based on the Somalia Integrated Household Budget Survey (SIHBS) data, 12.7% of children aged 6–18 had no formal education (SE = 0.0032; 95% CI: 12.0%– 13.3%). In contrast, 87.3% had attended school at some level (SE = 0.0032; 95% CI: 86.7%–88.0%). The narrow confidence intervals for both categories (spanning approximately 1.3 percentage points) indicate high precision in these national estimates.

### Health conditions and education attainment

Table 2 reveals consistent health disparities between educated and non-educated Somali children. Those without formal education exhibited higher rates of chronic illness (2.18% vs 0.97%), recent illnesses (4.21% vs 2.63%), and activity limitation days (23.6% vs 23.4%), while showing marginally better medical treatment access (2.93% vs 1.99%). The most striking finding remains the critically low healthcare utilization across both groups (<3% treatment rates), overshadowing the modest educational differences in health outcomes.

These patterns suggest that while health disadvantages may compound educational exclusion, Somalia's systemic healthcare failures affect all children regardless of schooling status. The near-identical rates of activity limitation days (despite higher illness prevalence among non-educated children) imply that functional health impacts may be mediated by factors like poverty or regional disparities rather than education alone. This reinforces the need for health system strengthening alongside targeted school-based interventions to break the health-education poverty trap.

### Descriptive and bivariate results of control variables

The descriptive analysis of 10,511 Somali children aged 6–18 revealed notable demographic and health-related distributions, as detailed in Table 3. Regionally, Banadir had the highest representation (9.7%), while Middle Shabelle had the lowest (2.3%). A majority of children resided in urban areas (69.1%), with smaller proportions in rural (26.0%) and

**Table 1. Prevalence of education attainment in Somalia.**

| Category | Proportion | Standard Error | 95% Confidence Interval |
|---|---|---|---|
| **No School Attainment** | 0.1265341 | 0.0032427 | 0.1203132 - 0.1330281 |
| **Attended School** | 0.8734659 | 0.0032427 | 0.8669719 - 0.8796868 |

**Table 2. Health conditions and education attainment.**

| | School attainment | | |
|---|---|---|---|
| **Chronic illness** | **No (1,330)** | **Yes (9,181)** | **Total** |
| Yes | 29 | 89 | 118 |
| No | 1,301 | 9,092 | 10,393 |
| Total | 1,330 | 9,181 | 10,511 |
| **Illness in the last month** | | | |
| Yes | 56 | 241 | 297 |
| No | 1,274 | 8,940 | 10,214 |
| Total | 1,330 | 9,181 | 10,511 |
| **Received medical treatment** | | | |
| Yes | 39 | 183 | 222 |
| No | 1,291 | 8,998 | 10,289 |
| Total | 1,330 | 9,181 | 10,511 |
| **Days Unable to Do Normal Activity** | | | |
| Yes | 314 | 2,145 | 2,459 |
| No | 1,014 | 7,020 | 8,034 |
| Total | 1,330 | 9,181 | 10,511 |

nomadic (4.9%) settings. The sample was nearly balanced by sex (51.3% male, 48.7% female) and across age groups. Most children lived with their mothers (82.8%) and fathers (63.0%), with a plurality coming from large families (7 + members). A small fraction of the sample were Internally Displaced Persons (IDPs) (8.0%), and just over half (52.7%) were classified as living in poverty. Health indicators showed low prevalence of chronic illness (1.1%) and recent illness (2.8%).

Bivariate analysis using chi-square tests, presented in Table 4, identified statistically significant predictors of school non-attendance ($p < 0.025$). Geographic disparities were the most pronounced (Region: $\chi^2 = 279.36$, $p < 0.001$), with non-attendance rates as high as 30.5% in Middle Shabelle. Residence type was also highly significant ($\chi^2 = 184.22$, $p < 0.001$), revealing that nomadic children faced triple the risk of non-attendance compared to their urban or rural peers. Age showed a steep and highly significant trend ($\chi^2 = 623.14$, $p < 0.001$), with non-attendance rising from just 6.1% for 6–10 year-olds to 24.6% for 15–18 year-olds.

Parental absence was significantly associated with higher non-attendance (Mother Live in Household: $\chi^2 = 30.30$, $p < 0.001$; Father Live in Household: $\chi^2 = 28.62$, $p < 0.001$). Health factors demonstrated selective but significant impacts: Chronic Illness ($\chi^2 = 15.35$, $p < 0.001$) more than doubled the risk of non-attendance, while a Recent Illness ($\chi^2 = 10.64$, $p = 0.001$) also significantly increased it. Notably, IDP Status ($\chi^2 = 14.34$, $p < 0.001$) was a significant predictor, while household Poverty Status ($\chi^2 = 1.91$, $p = 0.167$) was not statistically significant. Household size and days unable to do work showed no association.

## Bivariate analysis

The chi-square in Table 4 analysis identified several statistically significant predictors of school non-attendance (all $p < 0.025$). Geographic disparities were most pronounced (Region: $\chi^2 = 279.36$, $p < 0.001$), with children in Middle Shabelle (30.5% non-attendance) and Gedo (27.5%) faring worst. Residence type ($\chi^2 = 184.22$, $p < 0.001$) revealed nomadic children had triple the non-attendance of urban/rural peers. Age effects ($\chi^2 = 623.14$, $p < 0.001$) showed a steep dropout pattern from 6% (6-10y) to 25% (15-18y). Parental absence also significantly increased non-attendance (Mother: $\chi^2 = 30.30$, $p < 0.001$; Father: $\chi^2 = 28.62$, $p < 0.001$). Health factors showed selective impacts: chronic illness ($\chi^2 = 15.35$, $p < 0.001$) doubled non-attendance risk, while recent illness ($\chi^2 = 10.64$, $p = 0.001$) increased it by 6 percentage points. Notably, IDP status ($\chi^2 = 14.34$, $p < 0.001$) had stronger effects than poverty ($\chi^2 = 1.91$, $p = 0.167$, NS).

**Table 3. Descriptive analysis of control variables.**

| Variable and Categories | Frequency | Percentage |
|---|---|---|
| **Region** | | |
| Awdal | 443 | 4.21 |
| Bakool | 325 | 3.09 |
| Banadir | 1,018 | 9.69 |
| Bari | 720 | 6.85 |
| Bay | 568 | 5.40 |
| Galgaduud | 526 | 5.00 |
| Gedo | 586 | 5.58 |
| Hiraan | 563 | 5.36 |
| Lower Juba | 363 | 3.45 |
| Lower Shabelle | 583 | 5.55 |
| Waqooyi Galbeed | 996 | 9.48 |
| Middle Shabelle | 239 | 2.27 |
| Mudug | 640 | 6.09 |
| Nugaal | 777 | 7.39 |
| Sanaag | 690 | 6.56 |
| Sool | 707 | 6.73 |
| Togdheer | 767 | 7.30 |
| **Residence** | | |
| Rural | 2,731 | 25.98 |
| Urban | 7,264 | 69.11 |
| Nomadic | 516 | 4.91 |
| **Sex of Child** | | |
| Male | 5,394 | 51.32 |
| Female | 5,117 | 48.68 |
| **Age of Child** | | |
| 6-10 years (Primary) | 3,570 | 33.96 |
| 11-14 years (Middle School) | 3,665 | 34.87 |
| 15-18 years (High School) | 3,276 | 31.17 |
| **Mother Live in the Household** | | |
| Yes | 8,700 | 82.77 |
| No | 1,811 | 17.23 |
| **Father Live in the Household** | | |
| Yes | 6,623 | 63.01 |
| No | 3,888 | 36.99 |
| **Household Members** | | |
| 1-3 Persons (Small Family) | 1,198 | 11.40 |
| 4-6 Prsons (Medium Family) | 4,349 | 41.38 |
| 7 and more Persons (Large Family) | 4,964 | 47.23 |
| **IDP Status** | | |
| Yes | 838 | 7.97 |
| No | 9,673 | 92.03 |
| **Poverty Status** | | |
| Poor | 5,544 | 52.74 |
| Non poor | 4,967 | 47.26 |

*(Continued)*

**Table 3.** (Continued)

| Variable and Categories | Frequency | Percentage |
|---|---|---|
| **Chronic Illness** | | |
| Yes | 118 | 1.12 |
| No | 10,393 | 98.88 |
| **Illnesses in the last Month** | | |
| Yes | 297 | 2.83 |
| No | 10,214 | 97.17 |
| **Received Medical Treatment** | | |
| Yes | 222 | 2.11 |
| No | 10,289 | 97.89 |
| **Days unable to Do Work** | | |
| O days | 2,459 | 23.39 |
| 1-7 days | 8,034 | 76.43 |
| 8-30 days | 18 | 0.17 |

Several expected predictors failed to reach significance at $p < 0.025$. Household size ($\chi^2 = 1.53$, $p = 0.466$) and activity limitation days ($\chi^2 = 0.08$, $p = 0.963$) showed no association. Medical treatment access approached significance ($\chi^2 = 4.96$, $p = 0.026$), while sex differences ($\chi^2 = 7.14$, $p = 0.008$) were marginal. The lack of poverty effects ($\chi^2 = 1.91$, $p = 0.167$) contrasts with global trends, suggesting Somalia's educational exclusion is driven more by institutional and geographic barriers than household economics. These results highlight the need for targeted regional policies and health-education integration, particularly for nomadic populations and adolescents, rather than broad poverty-alleviation approaches.

## Machine learning models

To move beyond simple associations and identify the most powerful predictive factors, ten supervised machine learning models were trained to predict school attendance. As shown in Table 5 and the AUC-ROC curves in Fig 1, the Random Forest model emerged as the top performer with an AUC of 0.86 and a high specificity of 0.90. This indicates the model is particularly effective at correctly identifying children who have not attained schooling. While other models like K-Nearest Neighbors (KNN) showed higher sensitivity (0.77), Random Forest provided the best overall balance of performance metrics. Given its superior predictive power and balanced performance, the Random Forest model was selected for in-depth analysis to determine the hierarchy of barriers to school attainment.

## Feature importance

The feature importance analysis from the top-performing Random Forest model reveals a clear and compelling hierarchy of determinants for school attainment, as illustrated in Fig 2. Structural and demographic factors overwhelmingly emerged as the most powerful predictors. Age of Child was the single most influential feature, underscoring that a child's developmental stage is pivotal to their educational outcome in Somalia. This was followed closely by Region and Residence, highlighting the profound impact of geographic disparities and livelihood (urban, rural, or nomadic) on a child's access to education. Household-level factors, including Household Size and Poverty Status, were also identified as important predictors, though their influence was secondary to the primary structural drivers. The Sex of the Child and parental presence (Father Live in Household, Mother Live in Household) were moderately influential. Notably, direct health indicators such as Chronic Illness and Recent Illness ranked lower in predictive power than these structural and household factors. The minimal influence of variables like medical treatment received further underscores that in this context, acute health events are

**Table 4. Bivariate analysis of control variables.**

| Variable and Categories | School Attendance | | Chi-square (X²) | P-value |
|---|---|---|---|---|
| | **No** | **Yes** | | |
| **Region** | | | 279.3634 | 0.000 |
| Awdal | 48 (10.84) | 395 (89.16) | | |
| Bakool | 60 (18.46) | 265 (81.54) | | |
| Banadir | 151 (14.83) | 867 (85.17) | | |
| Bari | 87 (12.08) | 633 (87.92) | | |
| Bay | 95 (16.73) | 473 (83.27) | | |
| Galgaduud | 48 (9.13) | 478 (90.87) | | |
| Gedo | 161 (27.47) | 425 (72.53) | | |
| Hiraan | 48 (8.53) | 515 (91.47) | | |
| Lower Juba | 57 (15.70) | 306 (84.30) | | |
| Lower Shabelle | 79 (13.55) | 504 (86.45) | | |
| Waqooyi Galbeed | 90 (9.04) | 906 (90.96) | | |
| Middle Shabelle | 73 (30.54) | 166 (69.46) | | |
| Mudug | 57 (8.91) | 583 (91.09) | | |
| Nugaal | 66 (8.49) | 711 (91.51) | | |
| Sanaag | 58 (8.41) | 632 (91.59) | | |
| Sool | 68 (9.62) | 639 (90.38) | | |
| Togdheer | 84 (10.95) | 683 (89.05) | | |
| **Residence** | | | 184.2164 | 0.000 |
| Rural | 287 (10.51) | 2,444 (89.49) | | |
| Urban | 879 (12.10) | 6,385 (87.90) | | |
| Nomadic | 164 (31.78) | 352 (68.22) | | |
| **Sex of Child** | | | 7.1411 | 0.008 |
| Male | 637 (11.81) | 4,757 (88.19) | | |
| Female | 693 (13.54) | 4,424 (86.46) | | |
| **Age of Child** | | | 623.1448 | 0.000 |
| 6-10 years (Primary) | 218 (6.11) | 3,352 (93.89) | | |
| 11-14 years (Middle School) | 306 (8.35) | 3,359 (91.65) | | |
| 15-18 years (High School) | 806 (24.60) | 2,470 (75.40) | | |
| **Mother Live in the Household** | | | | |
| Yes | 1,030 (11.84) | 7,670 (88.16) | 30.2965 | 0.000 |
| No | 300 (16.57) | 1,511 (83.43) | | |
| **Father Live in the Household** | | | 28.6236 | 0.000 |
| Yes | 750 (11.32) | 5,873 (88.68) | | |
| No | 580 (14.92) | 3,308 (85.08) | | |
| **Household Members** | | | 1.5251 | 0.466 |
| 1-3 Persons (Small Family) | 160 (13.36) | 1,038 (86.64) | | |
| 4-6 Prsons (Medium Family) | 531 (12.21) | 3,818 (87.79) | | |
| 7 and more Persons (Large Family) | 639 (12.87) | 4,325 (87.13) | | |
| **IDP Status** | | | 14.3429 | 0.000 |
| Yes | 141 (16.83) | 697 (83.17) | | |
| No | 1,189 (12.29) | 8,484 (87.71) | | |
| **Poverty Status** | | | 1.9081 | 0.167 |
| Poor | 678 (12.23) | 4,866 (87.77) | | |
| Non poor | 652 (13.13) | 4,315 (86.87) | | |

*(Continued)*

**Table 4.** (Continued)

| Variable and Categories | School Attendance | | Chi-square (X²) | P-value |
|---|---|---|---|---|
| | **No** | **Yes** | | |
| **Chronic Illness** | | | 15.3494 | 0.000 |
| Yes | 29 (24.58) | 89 (75.42) | | |
| No | 1,301(12.52) | 9,092 (87.48) | | |
| **Illnesses in the last Month** | | | 10.6362 | 0.001 |
| Yes | 56 (18.86) | 241 (81.14) | | |
| No | 1,274 (12.47) | 8,940 (87.53) | | |
| **Received Medical Treatment** | | | 4.9553 | 0.026 |
| Yes | 39 (17.57) | 183 (82.43) | | |
| No | 1,291 (12.55) | 8,998 (87.45) | | |
| **Days unable to Do Work** | | | 0.0761 | 0.963 |
| O days | 314 (12.77) | 2,145 (87.23) | | |
| 1-7 days | 1,014 (12.62) | 7,020 (87.38) | | |
| 8-30 days | 2 (11.11) | 16 (88.89) | | |

**Table 5. ML model comparison.**

| Model (n = 10,511) | Confusion Matrix | Actual | | AUC | F1 Score | Balanced Accuracy | Sensitivity | Specivity |
|---|---|---|---|---|---|---|---|---|
| | **Predicted** | **No** | **Yes** | | | | | |
| **Random Forest** | No | 165 | 172 | 0.86 | 0.54 | 0.76 | 0.62 | 0.90 |
| | Yes | 101 | 1664 | | | | | |
| **Logistic** | No | 181 | 642 | 0.70 | 0.33 | 0.66 | 0.68 | 0.65 |
| | Yes | 85 | 1194 | | | | | |
| **XGBoost** | No | 109 | 129 | 0.81 | 0.43 | 0.66 | 0.40 | 0.92 |
| | Yes | 157 | 1707 | | | | | |
| **SVM** | No | 102 | 1329 | 0.69 | 0.12 | 0.32 | 0.38 | 0.27 |
| | Yes | 164 | 507 | | | | | |
| **KNN** | No | 205 | 185 | 0.83 | 0.62 | 0.83 | 0.77 | 0.89 |
| | Yes | 61 | 1651 | | | | | |
| **Neural Network** | No | 174 | 461 | 0.77 | 0.38 | 0.70 | 0.65 | 0.74 |
| | Yes | 92 | 1375 | | | | | |
| **Probit** | No | 182 | 672 | 0.70 | 0.32 | 0.65 | 0.68 | 0.63 |
| | Yes | 84 | 1164 | | | | | |
| **Decision Tree** | No | 145 | 290 | 0.75 | 0.41 | 0.69 | 0.54 | 0.84 |
| | Yes | 121 | 1546 | | | | | |
| **Naïve Bayes** | No | 183 | 649 | 0.69 | 0.33 | 0.66 | 0.68 | 0.64 |
| | Yes | 83 | 1187 | | | | | |
| **GBM** | No | 143 | 369 | 0.74 | 0.36 | 0.66 | 0.53 | 0.79 |
| | Yes | 123 | 1467 | | | | | |

less decisive than the chronic, systemic barriers a child faces. These findings strongly suggest that educational exclusion in Somalia is more fundamentally driven by where a child lives, how old they are, and their family's mode of existence, rather than by their individual health status or household wealth alone.

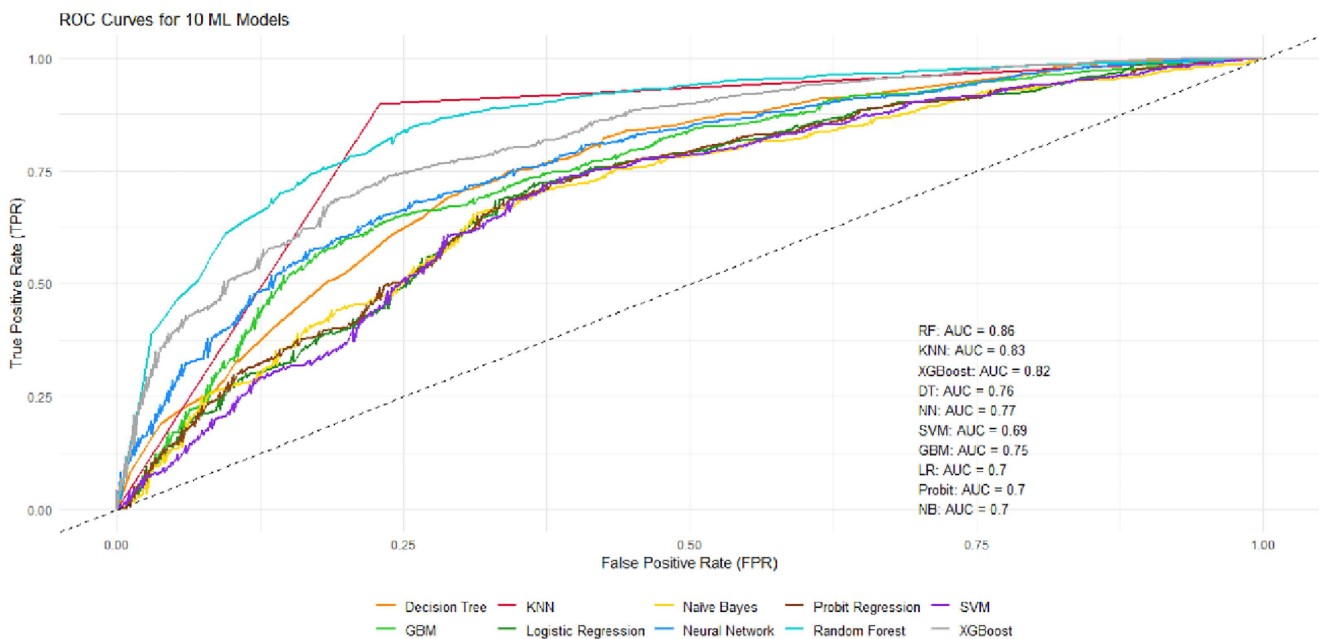

**Fig 1. AUC ROC curve.**

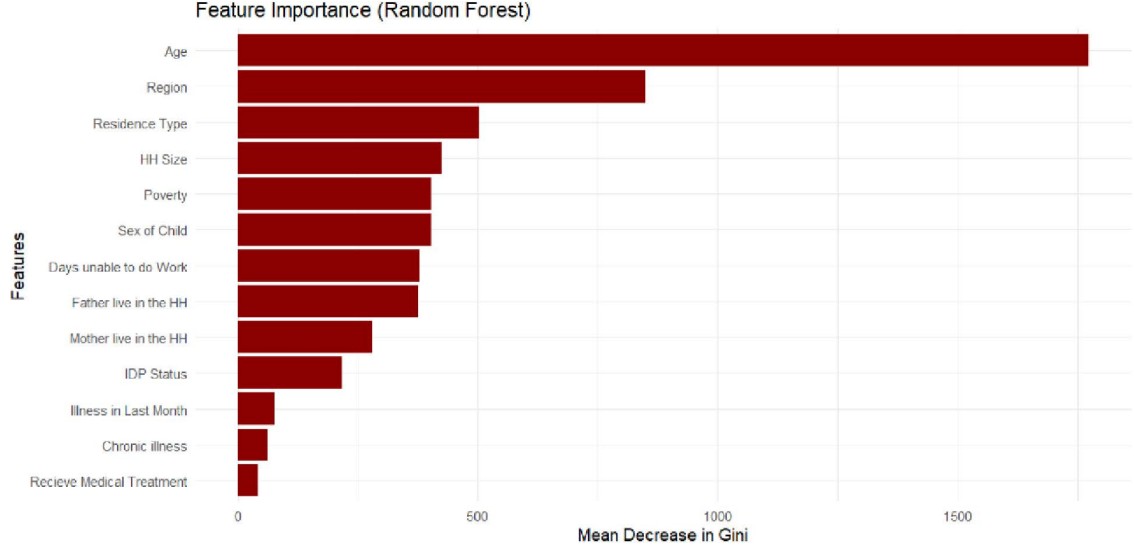

**Fig 2. Random forest feature importance.**

## Discussion

This study employed a machine learning methodology to identify the primary barriers to educational attainment in Somalia, revealing a hierarchy of factors that challenges conventional understanding. The central finding indicates that structural and demographic determinants—such as age, geographic region, and type of residence—are significantly more potent predictors of school attendance than individual health status or household poverty. Our Random Forest

model, characterized by high specificity (0.90), demonstrates that a child's location and life stage constitute the foremost barriers to their education. This finding is consistent with Bronfenbrenner's Ecological Systems Theory, which suggests that macrosystem and exosystem factors can exert a more profound influence on developmental outcomes than individual-level attributes, particularly in contexts of systemic fragility [28,31,32]. Age emerged as the strongest predictor of non-attendance. Our bivariate analysis revealed a dramatic increase in exclusion, from 6% in the 6–10 age group to nearly 25% among 15–18 year-olds. This finding corroborates a well-documented reality in many LMICs, where the risk of dropout escalates as children grow older due to rising opportunity costs, child labor, and early marriage [9]. Specific health challenges, such as inadequate menstrual hygiene management, also disproportionately affect adolescent girls' ability to remain in school [6,39].

The overwhelming importance of age in our model suggests that these combined pressures create a near-deterministic pathway out of education for many Somali adolescents. The profound impact of region and residence underscores Somalia's fragmented service delivery and the deep inequalities wrought by prolonged conflict. Children in insecure regions like Middle Shabelle and Gedo, or those from nomadic communities, face significantly higher rates of non-attendance. This empirically validates literature on the collapse of state infrastructure and the "weaponization" of services in conflict zones [10–12]. The high ranking of these geographic variables in our model suggests they act as proxies for a host of unobserved factors, including insecurity, the physical absence of schools, and weak governance, effectively creating educational "deserts" [14,44,65]. A critical insight from our analysis is the nuanced role of health. While bivariate tests confirmed that chronic and recent illnesses are significantly associated with non-attendance, their lower ranking in the ML model's feature importance hierarchy demands careful interpretation. This does not dismiss the health-education nexus but rather contextualizes it within a failed system [2,12]. In a nation where access to medical treatment is critically low for all children, as our sample confirms, an individual's health status loses its differential predictive power. When healthcare systems collapse systemically, poor health becomes a pervasive condition rather than a distinguishing characteristic of the out-of-school population [12,13]. The structural factors of region and residence thus subsume these health effects.

The interpretation is further corroborated by the non-significant direct effect of poverty status observed in our bivariate analysis and its mid-level importance in the machine learning (ML) model. This finding stands in stark contrast to the global literature, which frequently positions household poverty as a central factor in educational exclusion. In the Somali context, where over half of the population lives in poverty, poverty functions more as a constant than a variable. Our model illustrates that factors such as internally displaced person (IDP) status and geographic marginalization serve as more potent indicators of acute disadvantage. This observation aligns with intersectionality theory, which posits that overlapping identities related to displacement and location create barriers that transcend simple income-based poverty metrics [13,34]. Household-level factors, including parental presence, emerged as moderately influential predictors. The absence of a mother or father significantly elevates the risk of non-attendance, supporting Human Capital Theory's emphasis on parental investment in children's futures [24]. In a precarious environment, the stability and social capital provided by both parents are crucial for navigating the complex barriers to schooling. This finding reinforces the notion that household resilience acts as a critical mediator between macro-level shocks and individual educational outcomes. The moderate importance of a child's sex suggests that while gender disparities persist, they are interwoven with dominant structural factors. Girls in Somalia encounter unique and severe barriers, including gender-based violence and the challenges of menstrual health and hygiene (MHH) [6,34].

However, our ML model indicates that the disadvantages associated with living in a conflict-affected region or being an adolescent are stronger overall determinants of school attendance for both boys and girls. This calls for an intersectional approach that considers how gendered vulnerabilities are amplified by geography and life-cycle stage. The significance of IDP status as a predictor, ranking higher than poverty, underscores the profound disruption caused by displacement. As documented in studies of Somali IDP camps, displaced populations face a "triple jeopardy" of lost homes, severed community ties, and collapsed access to services [17,21]. The findings of [66] demonstrate this pattern across multiple

conflict-affected countries. Our model confirms that being an IDP constitutes a distinct and powerful barrier to education, independent of poverty, reinforcing the need for targeted programs for this uniquely vulnerable group of children. Methodologically, this study's application of machine learning makes a significant contribution. Unlike traditional regression models that can show multiple significant factors simultaneously, the Random Forest model's feature importance hierarchy provided a clear, data-driven ranking of determinants [55]. This approach follows a growing trend of leveraging ML to analyze complex health and social data [54,56] and has been successfully applied in the Somali context [22]. The model's high specificity (0.90) provides confidence in its ability to reliably identify out-of-school children, a crucial feature for effectively targeting limited resources.

The limited significance of variables such as 'days unable to work' and 'access to medical treatment' underscores the central theme of this study. The negligible influence of the former may be attributed to measurement challenges or the possibility that functional limitations are overshadowed by more immediate barriers to school attendance. The minimal impact of the latter starkly reflects a deteriorated health system; when treatment is largely inaccessible, its availability or lack thereof cannot meaningfully predict other outcomes [12]. Our findings collectively indicate that educational exclusion in Somalia is primarily driven by structural, geographic, and life-cycle factors. While the health-education nexus is real, it operates within these broader determinants. A child's health is undoubtedly important, yet their likelihood of attending school is more profoundly influenced by their age and geographic location in a country characterized by significant spatial and developmental disparities. This reframes the policy challenge from merely providing health interventions to dismantling systemic, geographically concentrated exclusion. In conclusion, this study validates the core tenets of Human Capital Theory by illustrating how external conditions constrain educational investments, while also highlighting the theory's limitations without the multi-level context provided by ecological frameworks [24,28]. By employing machine learning to empirically test the interactions proposed by these theories, our analysis offers a prioritized roadmap for action. It clarifies that to make education accessible for all Somali children, we must first address the structural divides that leave many behind.

## Policy implications

Our findings suggest three strategic policy shifts. First, educational interventions must be geographically targeted. Uniform national strategies are inadequate. The significant importance of region and residence necessitates a new focus. Resources should be concentrated in educational "coldspots" such as Gedo and Middle Shabelle. Support for nomadic communities is also crucial. Concrete actions are required, including the construction of secure schools and the deployment of mobile education units for pastoralist children. Additionally, robust incentives must be created to attract and retain qualified teachers in these underserved areas. Second, policies need an age-differentiated approach. The risk of dropout clearly escalates as children age. Universal primary education is insufficient. Robust programs must support the transition to secondary school. Adolescents require integrated services, which means combining education with essential health support, such as menstrual health provisions in schools. Flexible learning pathways are also necessary to accommodate the economic pressures faced by this age group. The goal is a resilient system that withstands the challenges of adolescence. Third, humanitarian actors should reconsider vulnerability. Simple poverty metrics are inadequate. IDP status and geographic location are stronger predictors of exclusion. Broad cash transfers may be less effective than targeted programs. Interventions must address the root causes of displacement and regional neglect. The focus should be on restoring services in marginalized regions. Comprehensive support for IDP families is essential, integrating education, health, and protection. Strengthening local governance is key to rebuilding the social contract.

## Conclusion

This study examined the intersection of health and education in Somalia, a nation where prolonged conflict has severely eroded human capital. Our machine learning analysis has uncovered a stark reality: the primary barriers to education are not individual health or wealth, but rather structural factors such as a child's age, region, and lifestyle.

These elements engender significant inequalities, ultimately determining a child's educational opportunities. This research makes a significant contribution by offering a data-driven, ranked understanding of educational exclusion, demonstrating that age, region, and residence are the most potent predictors of non-attendance. This shifts the focus from a generalized health-education link to specific structural challenges. The relatively lower predictive power of health and poverty is indicative of the context of systemic collapse, where macro-level factors overshadow individual outcomes. This finding carries profound policy implications. Methodologically, this study highlights the efficacy of machine learning in navigating complexity and extracting clear insights from extensive survey data. Our Random Forest model identified the most critical features among numerous variables, providing clarity that is often elusive with traditional methods. The result is robust evidence for prioritizing interventions where they are most needed. Ultimately, this research advocates for a novel approach. Advancing education in Somalia necessitates a targeted, structurally-aware strategy. Stakeholders must transcend individual factors and address the broader forces of inequality. Interventions should concentrate on the specific regions, age groups, and communities most disadvantaged. This approach is essential for constructing a more equitable and resilient educational system, one that ensures every Somali child has the opportunity to learn, thrive, and contribute to a better future.

## Study limitations

Despite its robust findings, this study has several limitations. Firstly, the cross-sectional nature of the SIHBS data implies that while we can identify strong predictors and associations, we cannot establish causality. It is not possible to definitively ascertain whether chronic illness leads to school non-attendance or if the factors associated with non-attendance also result in poorer health outcomes. Longitudinal data would be necessary to disentangle these complex causal pathways over time. Secondly, the analysis is constrained by the variables available in the dataset. The survey relies on self-reported data for health conditions and household characteristics, which may be subject to recall or social desirability bias. Furthermore, the dataset lacks detailed school-level data, such as teacher quality, infrastructure conditions, or proximity to the nearest school. The omission of these critical supply-side factors means our model, while powerful, is incomplete. The machine learning model can identify that a region is a strong predictor, but it cannot elucidate which specific community- or school-level factors within that region are driving the outcome.

## Recommendations for future research

Building upon these findings and limitations, two primary directions for future research are proposed. Firstly, there is a distinct need for mixed-methods and longitudinal research to enhance the depth and causal understanding of our predictive findings. Qualitative studies, including focus groups and in-depth interviews, should be conducted in the identified "coldspot" regions to elucidate the specific mechanisms and lived experiences underlying the quantitative data. Monitoring the educational and health trajectories of a cohort of children over time would facilitate a more rigorous analysis of causal relationships. Secondly, future research should concentrate on integrating household survey data with other data sources to enable a more comprehensive, multi-level analysis. Linking the SIHBS data with school-level data from Education Management Information Systems (EMIS), geospatial data on conflict events and climate shocks, and data on health facility locations would be particularly beneficial. This approach would allow researchers to transcend regional proxies and model the direct effects of school quality, community security, and service availability on educational attainment, thereby providing more precise guidance for policymakers.

## Supporting information

### S1 File. Cleaned Data Education.

(CSV)

## Author contributions

**Conceptualization:** Jibril Abdikadir Ali, Mustafe Khadar Abdi, Hodo Abdikarim.

**Data curation:** Jibril Abdikadir Ali, Mukhtaar Axmed Cumar, Mustafe Khadar Abdi.

**Formal analysis:** Hodo Abdikarim.

**Methodology:** Jibril Abdikadir Ali, Mukhtaar Axmed Cumar, Hodo Abdikarim.

**Software:** Abdisalam Hassan Muse, Hodo Abdikarim.

**Supervision:** Abdisalam Hassan Muse.

**Visualization:** Mustafe Khadar Abdi, Hodo Abdikarim.

**Writing – original draft:** Jibril Abdikadir Ali, Mukhtaar Axmed Cumar, Abdisalam Hassan Muse, Hodo Abdikarim.

**Writing – review & editing:** Jibril Abdikadir Ali, Mukhtaar Axmed Cumar, Mustafe Khadar Abdi, Abdisalam Hassan Muse, Hodo Abdikarim.

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
