## [Editor Report · Decision Letter 0]

18 Jul 2025

PONE-D-25-31078Uncovering the Health-Education Nexus: A Machine Learning Analysis of School Attainment Barriers in SomaliaPLOS ONE

Dear Dr. Abdikarim,

Thank you for submitting your manuscript to PLOS ONE. After careful consideration, we feel that it has merit but does not fully meet PLOS ONE’s publication criteria as it currently stands. Therefore, we invite you to submit a revised version of the manuscript that addresses the points raised during the review process.

We look forward to receiving your revised manuscript.

Kind regards,

Abdulaziz Althewini

Academic Editor

PLOS ONE

Additional Editor Comments:

Dear Authors,

Thank you for submitting your manuscript to our journal. I commend you for addressing a deeply significant and under-explored topic—the intersection of health and education in Somalia—using both traditional and machine learning methodologies. The scope and ambition of your study are evident, and your integration of empirical modeling with social theory represents a meaningful contribution to the field.

After careful review, I offer the following suggestions aimed at strengthening the clarity, coherence, and impact of your manuscript. These comments are intended to be constructive and to support your efforts in making the paper more accessible, rigorous, and policy-relevant.

1. Title and Framing

While the current title is compelling, it emphasizes the health-education nexus in a way that may mislead readers, given that your findings ultimately show structural and geographic factors (e.g., age, region, and residence) are stronger predictors of school non-attendance than health-related variables. You may wish to consider refining the title to better align with this emphasis, for example by highlighting structural exclusion or educational inequality in fragile settings.

2. Abstract Clarity

The abstract is thorough and informative, but its density—especially the inclusion of technical terms and statistical metrics—may overwhelm readers unfamiliar with machine learning methods. I recommend simplifying and focusing the abstract around three key areas: the research gap, the main findings, and the policy implications. Statistics can be limited to only those essential to conveying impact.

3. Scope of the Literature Review

Your literature review is exceptionally comprehensive, drawing from a wide array of frameworks and sources. However, it at times becomes difficult to follow due to the sheer volume of content and theoretical references. Consider streamlining this section by grouping theories into thematic clusters (e.g., structural determinants, child development frameworks, access models) and more explicitly connecting them to your research question and variable selection.

4. Theoretical Application

You refer to several conceptual frameworks—such as Bronfenbrenner’s ecological model, Human Capital Theory, and the 3-Delays model—but the linkage between these frameworks and the operational aspects of your study (i.e., variable selection and machine learning inputs) could be clarified. A conceptual diagram or paragraph illustrating how these frameworks informed your analytical choices would help the reader better understand the structure of your argument.

5. Justification for Machine Learning Approach

The inclusion of ten machine learning models is a strength, but the rationale for selecting these specific algorithms, and how their performance aligns with the study’s goals, could be elaborated. Additionally, it would be helpful to briefly explain how you balanced accuracy with interpretability, and why the Random Forest model was ultimately selected for feature importance analysis.

6. Interpretation of Feature Importance

Your feature importance analysis is a valuable contribution. However, more nuanced interpretation of mid- and lower-ranked predictors—particularly the health variables—would strengthen the discussion. While health indicators rank lower in predictive power, they are not irrelevant. Further elaboration on why they might underperform in Somalia’s context (e.g., systemic health system collapse, data limitations) would provide richer insight.

7. Presentation of Results

The manuscript presents a wealth of quantitative information, but could benefit from more visual representation. The addition of well-labeled and concise figures (e.g., ROC curves, feature importance charts, regional non-attendance heatmaps) would enhance accessibility and support your key findings. In-text references to tables and figures should also be clearer and more consistent.

8. Policy Relevance

Your policy implications are thoughtful and evidence-based. However, they could be made more concrete. For example, rather than general calls for geographically targeted interventions, you might suggest specific modalities—such as mobile schooling units for nomadic populations, or adolescent-focused health-education integration in regions like Gedo and Middle Shabelle.

9. Methodological Transparency

While the analytical methods are well-structured, a brief summary of preprocessing steps (e.g., treatment of missing data, variable encoding, model tuning strategies) would enhance reproducibility. A short methodological appendix or table outlining these details would be sufficient and beneficial.

10. Style, Grammar, and Formatting

There are some typographical errors and formatting inconsistencies throughout the manuscript (e.g., “INTRODUCITON,” uneven spacing in tables). Paragraphs in several sections are quite long, which can challenge readability. A thorough copy edit to improve language clarity, punctuation, and structural flow is strongly advised.

This is an important and original study that uses cutting-edge methodology to inform a pressing development issue. With some refinement to the structure, theoretical grounding, and presentation, your manuscript can significantly advance scholarly and policy understanding of educational exclusion in Somalia and similar fragile settings. I appreciate the rigorous work you have put into this research and look forward to seeing a revised version that incorporates these suggestions.

---

## [Author Response · Author response to Decision Letter 1]

20 Jul 2025

Response to Reviewers

Dear Dr. Althewini and Reviewers,

We sincerely appreciate your thoughtful feedback on our manuscript, "Uncovering the Health-Education Nexus: A Machine Learning Analysis of School Attainment Barriers in Somalia." We have carefully addressed each of your comments and revised the manuscript accordingly. and we have attached the documents

We believe these revisions significantly strengthen the manuscript’s clarity, theoretical grounding, and policy impact. Thank you again for your constructive feedback—we are grateful for the opportunity to improve our work.

Sincerely,

Hodo Abdikarim (on behalf of all authors)

Attachments

1. Revised Manuscript with Track Changes

2. Clean Manuscript

3. Response to Reviewer

4. TIF Figures (1&2)

---

## [Decision Letter · Decision Letter 1]

30 Sep 2025

Beyond Health: A Machine Learning Analysis of Structural Barriers to School Attainment in Somalia

PONE-D-25-31078R1

Dear Dr. Abdikarim,

We’re pleased to inform you that your manuscript has been judged scientifically suitable for publication and will be formally accepted for publication once it meets all outstanding technical requirements.

Kind regards,

Abdulaziz Althewini

Academic Editor

PLOS ONE

Additional Editor Comments (optional):

Reviewers' comments:

Reviewer's Responses to Questions

**Comments to the Author**

1. If the authors have adequately addressed your comments raised in a previous round of review and you feel that this manuscript is now acceptable for publication, you may indicate that here to bypass the “Comments to the Author” section, enter your conflict of interest statement in the “Confidential to Editor” section, and submit your "Accept" recommendation.

Reviewer #1: (No Response)

2. Is the manuscript technically sound, and do the data support the conclusions?

Reviewer #1: Yes

3. Has the statistical analysis been performed appropriately and rigorously?

Reviewer #1: Yes

4. Have the authors made all data underlying the findings in their manuscript fully available?

Reviewer #1: Yes

5. Is the manuscript presented in an intelligible fashion and written in standard English?

Reviewer #1: Yes

6. Review Comments to the Author

Reviewer #1: (No Response)

7. PLOS authors have the option to publish the peer review history of their article (what does this mean? ). If published, this will include your full peer review and any attached files.

**Do you want your identity to be public for this peer review?** For information about this choice, including consent withdrawal, please see our Privacy Policy .

Reviewer #1: No

---

## [Editor Report · Acceptance letter]

PONE-D-25-31078R1

PLOS ONE

Dear Dr. Abdikarim,

I'm pleased to inform you that your manuscript has been deemed suitable for publication in PLOS ONE. Congratulations! Your manuscript is now being handed over to our production team.

Kind regards,

on behalf of

Dr. Abdulaziz Althewini

Academic Editor

PLOS ONE